# Action of Photodynamic Therapy at Low Fluence in 9 L/lacZ Cells after Interaction with Chlorins

**Gabrielle dos Santos Vitorio** [1], **Bruno Henrique Godoi** [1], **Juliana Guerra Pinto** [1,*], **Isabelle Ferreira** [1], **Cristina Pacheco Soares** [2] and **Juliana Ferreira-Strixino** [1]

1   Photobiology Applied to Health Lab (PhotoBioS), Research and Development Institute (IP & D), University of Vale do Paraíba (UNIVAP), Avenida Shishima Hifumi, 2911, Urbanova, São José dos Campos 12244-000, São Paulo, Brazil
2   Laboratório de Dinâmica de Compartimento Celulares, Research and Development Institute (IP & D), University of Vale do Paraíba (UNIVAP), Avenida Shishima Hifumi, 2911, Urbanova, São José dos Campos 12244-000, São Paulo, Brazil
*   Correspondence: juguerra@univap.br

**Abstract:** Gliosarcoma (GS) is a primary malignant neoplasm of the central nervous system, treated with an unfavorable prognosis with surgery, radiotherapy, and chemotherapy. The treatment for GS consists of surgical resection, almost always accompanied by radiotherapy and/or chemotherapy, given the invasive behavior of the tumor. Photodynamic Therapy (PDT) is studied as an alternative method that combines light, a photosensitizer (PS), and molecular oxygen. This study aimed to compare the effects of PDT using the photosensitizers Fotoenticine (FTC) and Photodithazine (PDZ) at low concentrations and fluences. For this study, 9 L/lacZ cells, concentrations of 1.55 µg mL$^{-1}$, 12.5 µg mL$^{-1}$, and 50 µg mL$^{-1}$ of chlorins and fluences of 1, 5, and 10 J/cm$^2$ were used. A test was also carried out with Trypan Blue in L929 cells at the mentioned concentrations at 5 J/cm$^2$. Both chlorins were internalized in the cytoplasm, with a significant reduction in viability (>95%) in almost all groups and altered cell adhesion and morphology after PDT. HSP70 expression decreased in both PS, while HSP27 increased only in PDT with FTC, and although there was a change in cell adhesion in the 9 L/LacZ lineage it was not observed in the L929 fibroblast lineage. Both chlorins were effective, highlighting the concentration of 50 µg mL$^{-1}$ at the fluence of 5 J/cm$^2$; according to the present study, the PDZ showed better results.

**Keywords:** gliosarcoma; brain cancer; photodynamic therapy; photodithazine; fotoenticine; heat shock protein



## 1. Introduction

Central Nervous System (CNS) cancer is one of the most aggressive with an unfavorable prognosis, with an expectation of more than 11,000 new cases for each year of the 2020–2022 triennium in Brazil, according to data published by the National Cancer Institute (INCA) [1]. Among the primary tumors of the CNS, gliosarcoma (GS) is a tumor of biphasic histological differentiation composed of a combination of neoplastic glial cells and mesenchymal elements. It is believed that the origin of gliosarcoma is due to sarcomatous transformation that can occur in some cases of glioblastomas, corresponding to between 2 and 8% of cases. It frequently appears in the temporal lobe, and is more frequent in males, mainly in the age group from 40 to 60. This type of tumor presents rapid progression and a tendency towards extracranial metastases and wild Isocitrate Dehydrogenase, characterizing its high degree of malignancy [1–3].

The treatment for GS consists of surgical resection, almost always accompanied by radiotherapy and/or chemotherapy. Considering the invasive behavior of the tumor, these adjuvant therapies are used to eliminate as much of the remaining tumor tissue as possible. With multimodal treatment, patients survive on average from 10 to 15 months, but the

therapy directly impacts their quality of life since systemic side effects can significantly weaken their health [1,4,5].

Photodynamic Therapy (PDT) has been evaluated in several diseases as a safe and selective treatment option to minimize undesirable side effects and use more effective methods to eradicate the tumor. Its principle consists of using a photosensitive substance, molecular oxygen, and light at a suitable wavelength. The tumor cell interacts with the photosensitizer (PS) and internalizes it in its cytoplasm. When light is emitted into tumor tissue, reactions occur between the PS and molecular oxygen, triggering the formation of reactive oxygen species (ROS), activating a cascade of events that can lead to cell death [6–8].

Therefore, researchers seek to find photosensitizers that have efficient irradiation parameters to significantly reduce the number of malignant cells with few treatment sessions. In this sense, chlorins have several positive points, such as absorption at wavelengths in the region of 660 nm, the capability of accumulating in deeper tumors, and low cost, compared to other compounds, such as phthalocyanines and PS carried by nanoparticles [9,10]. Fotoenticine (FTC) is a Spanish chlorine, synthetically produced by the company Nuevas Tecnologías Científicas—NTC and Photodithazine (PDZ) is a derivative of chlorin e6, supplied by the Russian company Veta-Grand® (Figure 1). Both chlorins already have publications that demonstrate promising results in use with PDT for the treatment of GS, encouraging the continuation of their studies as potential PS [3,11,12].

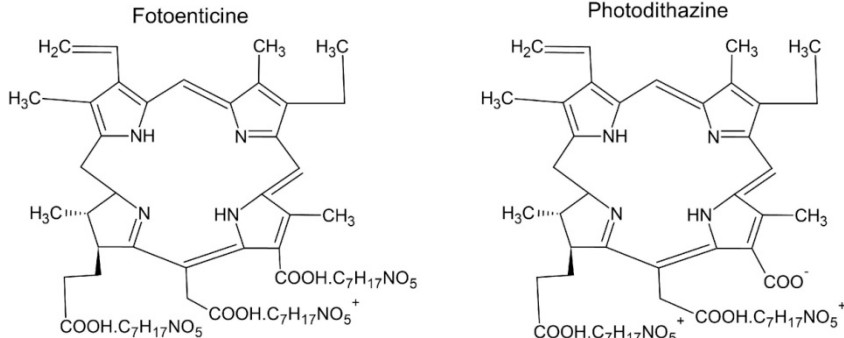

**Figure 1.** Chemical Structure of the Fotoenticine and Photodithazine.

In addition to significantly affecting the number of viable cells, it is essential to understand the effects triggered by PDT in glioblastoma cells, such as morphological changes, processes indicative of cell death, and changes in the expression of proteins in response to the stress suffered by the cell, as occurs in PDT [13,14], in addition to seeking the optimal dose for the safe application of PDT. Thus, the objective of this study was to analyze some of the effects triggered by PDT with the lower concentrations of FTC and PDZ in the treatment of GS cells, comparing the results obtained from both PS, and observing the cellular internalization, alteration in viability, adhesion, cell morphology, and protein expression after PDT.

## 2. Materials and Methods

### 2.1. Cell Lineage

Gliosarcoma cells derived from the brain tissue of rats (Rattus norvegicus) with fibroblastic morphology of lineage 9 L/lacZ purchased at the Cell bank from Rio de Janeiro, Brazil, (BCRJ® CRL-2200TM) and murine fibroblast cells originating from connective tissue lineage L929 supplied by Gibco® (Waltham, MA, USA, maintained in medium DMEM (Gibco) with F12 (Gibco) and supplemented with 10% Fetal Bovine Serum, 1% Penicillin/Streptomycin solution (LGC Biotecnologia, Cotia, Brazil), were placed in an incubator at 37 °C with 5% $CO_2$. The 9 L/lacZ cell line was obtained from Banco de Células do Rio de Janeiro—BCRJ, Rio de Janeiro, Brazil and the L929 cell line was provided by Adolfo Lutz Institute, São Paulo, Brazil.

## 2.2. Photosensitizer

The Spanish chlorine Fotoenticine (FTC), from the company Nuevas Tecnologías Científicas—NTC—Llanera (Asturias, Spain), and the Russian chlorine Photodithazine (PDZ), produced by Veta-Grand® (Moscow, Russia) were used. Commercially, they are available at 7 mg mL$^{-1}$ and 5 mg mL$^{-1}$, respectively. For this study, they were diluted in PBS (Phosphate-buffered saline) at concentrations of 50 µg·mL$^{-1}$, 12.5 µg·mL$^{-1}$, and 1.55 µg·mL [12,13]. PDZ and FTC were kept in the dark during the process and stored at 4 °C.

## 2.3. Light Source

An LED-based device Irrad-Led 660 (Biopdi, São Carlos, Brazil) was used at fluences of 1 J/cm$^2$, 5 J/cm$^2$, and 10 J/cm$^2$ and an irradiance of 25 mW/cm$^2$.

## 2.4. PS Internalization of PS by the 9 L/lacZ Lineage

In 24-well plates with round coverslips at the bottom, $1 \times 10^5$ cells were seeded and incubated for 24 h at 37 °C. Cells were incubated with PDZ and FTC for 1 h in an incubator at 37 °C. After this period, the PS was removed, the wells were washed once with PBS at room temperature, and fixed in solution of paraformaldehyde (Dinâmica Química Contemporânea LTDA, Indaiatuba, Brazil) at 4% for 10 min. Then, the paraformaldehyde was removed, and the wells were washed with PBS. The slides were mounted with one drop of Prolong with DAPI (4′,6-diamidino-2-phenylindole, dihydrochloride) (Invitrogen—Thermo Fisher Scientific, Waltham, MA, USA). All processing was carried out in the dark, and the slides were examined in a confocal microscope LSM 700 Zeiss with a 63× objective (DAPI λexc 405 nm, λem 461 nm; PS λexc 488 nm λ at above 500 nm). The images were mounted and the bar inserted in ZEISS Software ZEN Blue Edition 3.7.

## 2.5. Preparing for the Trypan Blue and MTT Assay

All assays were performed in triplicate, in 96-well plates, with a cell concentration of $1 \times 10^4$ cells/well. The dark group comprises the control and the PS at the concentrations studied without irradiation, allowing us to verify whether only the PS can trigger some cytotoxic process when in contact with the cells. The irradiated group is composed of a control group, without adding PS, and the groups corresponding to PDT. After an incubation period of 24 h, for cell adhesion, the PS were incubated for 1 h at 37 °C in the absence of light. Then, the medium was removed and replaced by PBS, and the groups were irradiated or kept in the dark, according to the group. After irradiation, the PBS was removed, and the culture medium was added. Viability tests were performed 18 h after irradiation.

## 2.6. Analysis of Cell Viability and Adherence by Exclusion Assay with Trypan Blue

The Trypan Blue Exclusion Assay is a cytotoxicity test that allows differentiating live from dead cells by observing the staining of the cell. Due to the integrity of the membrane, there is no dye accumulation in the cytoplasm of viable cells. This test was performed on both the 9 L/lacZ and L929 strains to observe the effects of PDT on malignant and normal cells. The cells that remained adhered to the wells were counted, and the cells that detached and remained in the supernatant. After 18 h of treatments, the culture medium of each well was transferred to Eppendorfs to analyze the presence of cells that lost adhesion during this period and their viability. The Eppendorfs were centrifuged at 1050 RCF for 10 min. The formed pellet was resuspended in 30 µL of 0.2% Trypan Blue solution (Sigma®, San Luis, MO, USA). After 5 min of incubation, 10 µL of the Eppendorf was added to a Neubauer chamber for cell counting. For cells adhered to the plate, 70 µL of 0.2% Trypan Blue solution was added to each well. After 5 min, the Trypan Blue solution was removed, and 100 µL of PBS was added. The wells were photographed using a camera coupled to an inverted optical microscope (Zeiss®—Axio Vert A1, Gottingen, Germany) at 40× magnification, with 5 random fields selected from each well. Dead cells (when blue staining was retained) and live cells (non-stained cells) of each group were counted using

ImageJ® version 1.51 software (Plugins—Analyze—Cell Counter). The analysis of changes in cell adherence after PDT was performed quantitatively and qualitatively through the images obtained.

### 2.7. Analysis of Mitochondrial Activity as a Complementary Test of Viability

Mitochondrial activity was evaluated using the MTT assay (3-(4,5-dimethylthiazol-2-yl)-2,5diphenyltetrazoliumbromide) (Invitrogen—Thermo Fisher Scientific, Waltham, MA, USA), consisting of the degradation of the MTT salt in Formazan crystals by viable cells. After 18 h of the treatments, 50 µL of MTT diluted in PBS (5 mg·mL$^{-1}$) was added to each well, gently shaken, and incubated for 3 h at 37 °C in a humidified atmosphere in the absence of light. After this period, the solution was removed, and 100 µL of DMSO (Dimethyl sulfoxide) (LGC Biotecnologia, Cotia, Brazil) was added to each well to solubilize the formazan crystals. Optical density was measured using an optical microplate reader (Biotek Synergy HT Spectrophotometer, Winooski, VT, USA) with a 570 nm filter.

### 2.8. Scanning Electron Microscopy (SEM)

For this test, the groups irradiated at a fluence of 5 J/cm$^2$ were selected. After 18 h of PDT, the cells were washed with PBS once and fixed with a solution of 2.5% glutaraldehyde (Dinâmica Química Contemporânea LTDA, Indaiatuba, Brazil), 4.0% paraformaldehyde (Dinâmica Química Contemporânea LTDA, Indaiatuba, Brazil), cacodylate buffer (Electron Microscopy Sciences, Hatfield, USA) 0.1 M, 1 mM Calcium Chloride (50 mM stock) (Sigma®, San Luis, MO, USA, and osmosis water. After this period, the cells were washed three times with cacodylate buffer (0.1 M) and post-fixed with Osmium (Electron Microscopy Sciences, Hatfield, EUA) for 30 min. Then, the cells were washed with cacodylate buffer and subsequently dehydrated with ethanol with the following sequence: 15 min with 70% alcohol, 15 min with 90% alcohol, and 15 min with 100% alcohol (repeated 4 times). After this process, a drop of HDMS (hexadimethyl disilazone) (Electron Microscopy Sciences, Hatfield, PA, USA) was added, and the metallization process was performed (Metalizadora Emitech K550X, Fall River, MA, USA). The images were captured using the MEV EVO MA 10 Zeiss Coupled EDX equipment from the Multiuser Lab Center, at the Vale do Paraíba Research and Development Institute (São José dos Campos, Brazil).

### 2.9. Extraction and Expression Analysis of Heat Shock Proteins (HSP) HSP27 and HPS 70

To the protein extraction, $1 \times 10^5$ cells were plated in 6-well plates, and cells were allowed to grow over the entire length of the wells, performing maintenance every 48 h. After this process, the cells were subjected to PDT and incubated for 18 h after treatment. Next, wells were washed with ice-cold PBS, and cells were scraped and transferred to labeled 15 mL tubes. The tubes were centrifuged for 10 min at 1050 RFC, the supernatant was discarded, and 1 mL of RIPA buffer solution was added, resuspended in the pellet, and incubated for 30 min at 4 °C in an ice bath. After incubation, the cells were centrifuged for 20 min at 4 °C in 15,777 RCF. Finally, the supernatant was transferred to Eppendorfs, and protein quantification was performed by the Bradford Method, measured with a filter at 570 nm by the spectrophotometer (SpectraCount Packard, Meriden, CT, USA). Protein quantification by ELISA: After measuring the protein concentration by the Bradford Method, the sensitization process was started with the antigens in 96-well plates. Thus, aliquots were separated at a concentration of 10 µg diluted in 0.1 m bicarbonate buffer pH = 9.0, and 100 µL of the solution was applied per well in two plates, under agitation for 5 min, and left overnight in a humid chamber at 4 °C. For the application of primary antibodies HSP27 and HSP70 [14], the plates were washed with PBS-Tween 0.02% twice and blocked with the PBSTL solution (PBS-Tween 0.02% (1000 mL) (Synth) + milk (3 g)), then the plates were shaken and incubated at 37 °C for 1 h. The plates were again washed with PBSTL, and the primary antibody diluted in PBSTL was applied, which was shaken and incubated in an oven at 37 °C for another 1 h. Subsequently, the plates underwent three more washes with PBSTL, and the secondary antibody was applied in PBSTL, which

was shaken and incubated for 1 h in an oven at 37 °C. Finally, the plates were washed three times with PBS-Tween 0.02%, and Ortho-phenylenediamine dihydrochloride (OPD) solution (Merck, Darmstadt, Germany) was applied for 30 min in a dark chamber. Subsequently, 0.1 M HCl was added to stop the reaction, and the reading was performed in a spectrophotometer (SpectraCount Packard) with a filter at 490 nm.

*2.10. Statistical Analysis*

The data from the abovementioned tests were submitted to the ANOVA test—One way (BioEstat 5.0, Intituto Mamirauá, Belém, Brazil), using $\alpha = 0.05$ as a significance level.

## 3. Results

*3.1. PS Cellular Internalization*

Using confocal fluorescence microscopy, the internalization of PS in the cytoplasmic extension was observed in all tested concentrations. The blue fluorescence corresponds to the DAPI signal, and it is bound to the cell's DNA in the nucleus. The excited PS emits the signal represented in red, observed in the cytoplasmic region, without interaction with the nucleus (Figure 2).

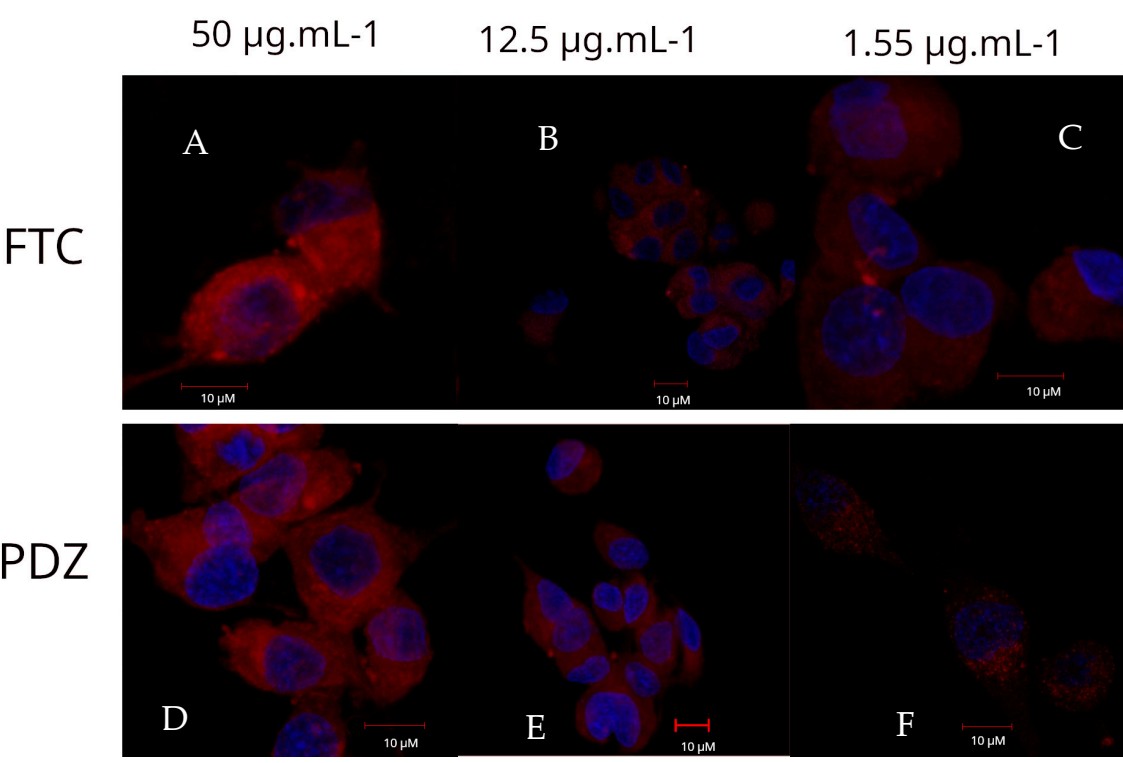

**Figure 2.** Internalization of PS in 9 L/lacZ cells by fluorescence confocal microscopy. (**A**)—50 μg mL$^{-1}$ FTC; (**B**)—μg mL$^{-1}$ FTC; (**C**)—1.55 μg mL$^{-1}$ FTC; (**D**)—50 μg mL$^{-1}$ PDZ; (**E**)—μg mL$^{-1}$ PDZ; (**F**)—1.55 μg mL$^{-1}$ PDZ. The DAPI labeling is observed in blue, attributed to the DNA, and the fluorescence of FTC and PDZ in red, in the cytoplasm.

*3.2. Cytotoxicity Test (Trypan Blue and MTT)*

Cytotoxicity tests showed similar responses in all light fluences tested with the concentrations of 50 μg·mL$^{-1}$ and 12.5 μg·mL$^{-1}$ of FTC and PDZ, with approximately 98% reduction in viability and more than 86% reduction in mitochondrial activity (Figures 3 and 4). For both PS, the concentration of 1.55 μg·mL$^{-1}$ at the fluence of 1 J/cm$^2$ did not present effective results as much as the other parameters, as less than a 0% reduction in cell viability was observed with FTC and less than a 35% reduction with PDZ.

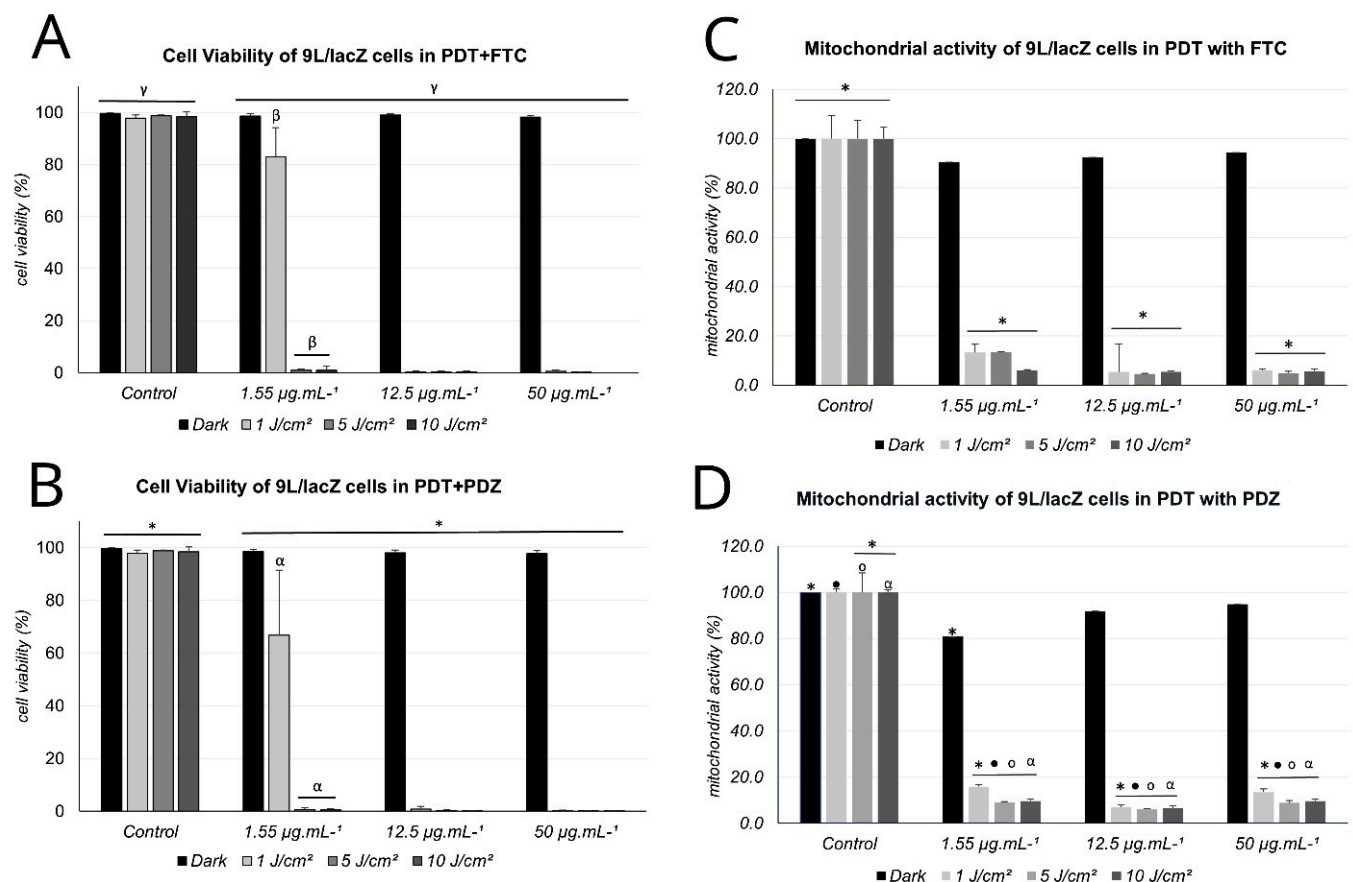

**Figure 3.** (**A**)—Cell viability evaluated by the exclusion test with Trypan Blue in the control and incubated with FTC, dark and irradiated groups. (**B**)—Cell viability evaluated by the Trypan Blue exclusion test in the control and PDZ-incubated, dark and irradiated groups. Symbols for $p < 0.01$: $\gamma$, $\beta$, * and $\alpha$. (**C**)—Mitochondrial activity evaluated by the MTT test in the control and incubated with FTC, dark and irradiated groups. (**D**)—Mitochondrial activity evaluated by the MTT test in the control and incubated with PDZ, dark and irradiated groups. Symbols for $p < 0.01$: *, ●, ○ and $\alpha$.

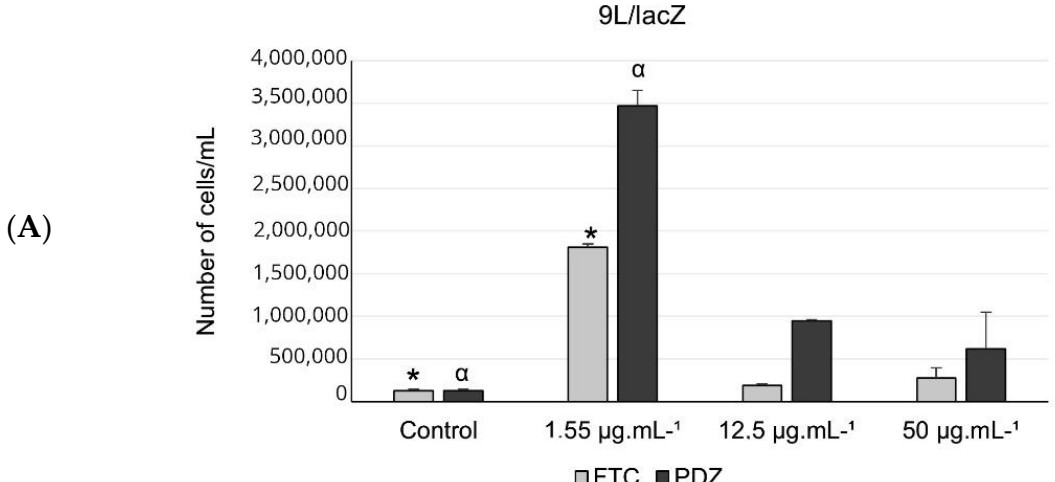

**Figure 4.** *Cont.*

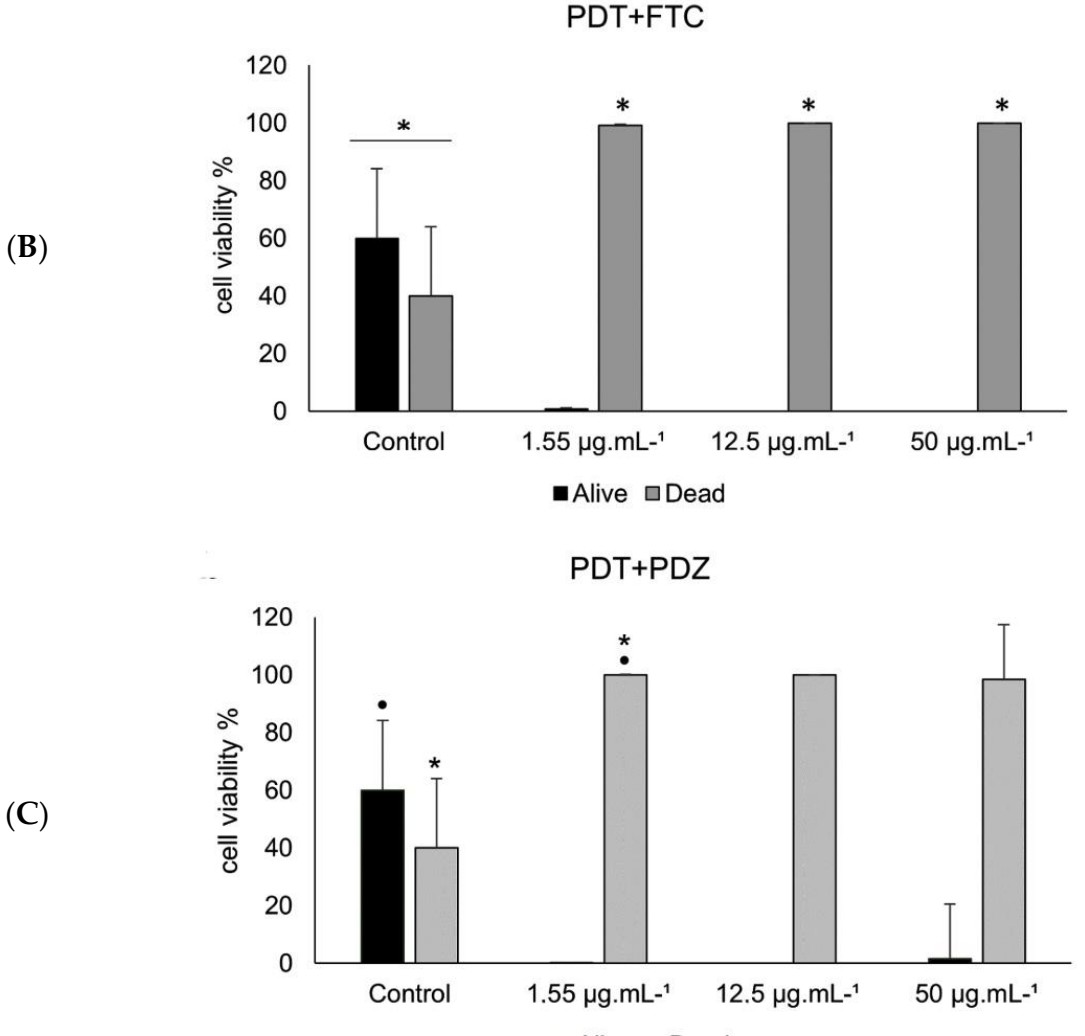

**Figure 4.** (**A**)—Number of cells present in the supernatant 18 h after PDT application at a fluence of 5 J/cm$^2$ in gliosarcoma cells with FTC and PDZ and control groups. Symbols for $p < 0.01$: * and α. (**B**)—Viability of the cells in the supernatant 18 h after PDT with FTC, and control group irradiated at 5 J/cm$^2$. (**C**)—Viability of the cells in the supernatant 18 h after PDT with PDZ, and control group irradiated at 5 J/cm$^2$. Symbols for $p < 0.01$: * and ●.

Despite this, the mitochondrial activity of the groups treated with both chlorins, irradiated or not, was reduced, demonstrating that even when the cells remained viable, as in the groups treated with 1.55 μg mL$^{-1}$ at the fluence of 1 J/cm$^2$, PDT could alter mitochondrial metabolism, reducing approximately 84% of its activity (Figure 3). Thus, the analysis of the results obtained with the MTT and Trypan Blue tests was similar at fluences of 5 J/cm$^2$ and 10 J/cm$^2$, indicating the effectiveness of PDT in reducing viable gliosarcoma cells. Since the lowest light fluence used that showed promising results in all concentrations was 5 J/cm$^2$, it was decided to continue using this fluence in the subsequent tests.

### 3.3. Change in Cell Adhesion and Viability Analysis of Supernatant Cells

In addition to evaluating the viability of the cells after PDT, the Trypan Blue test also allowed the visualization of changes in cell adhesion patterns after 18 h of treatment.

A discrepancy was observed in the number of adhered cells after treatment, compared with the seeded cells, and this reduction was inversely proportional due to the concentration in both PS. This fact was verified both qualitatively, through the visual analysis carried out during the execution of the experiments, and quantitatively, as shown in Figure 4.

The 12.5 µg·mL$^{-1}$ and 1.55 µg·mL$^{-1}$ concentrations showed a lower number of adhered cells per well than the higher concentration. However, when the supernatant cells were counted, we observed that the 1.55 µg·mL$^{-1}$ concentration had a more significant impact on cell detachment.

Cellular detachment characterizes a stress process suffered by the cell during PDT, which may represent the triggering of cell death or natural processes of the cell's life cycle. With that in mind, a test was performed to assess the ability of the cells in the supernatant to adhere again, transferring them to a new plate and observing for up to 72 h to verify if there would be adhesion. It was observed that only the control cells in the light group adhered again, as those that were submitted to PDT no longer had this capacity.

In addition, the viability was analyzed with the exclusion test with Trypan Blue in the supernatant cells, which were detached after PDT. It was observed that more than 90% of the cells were dead after the detachment triggered by PDT in both chlorins (Figure 4). Thus, we can assume that the process of cell detachment can be linked to the cascade of events triggered by cell death.

### 3.4. Analysis of Morphological Alteration by SEM

Using scanning electron microscopy (Figure 5A), in the group treated only with the light, the usual format of the 9 L/lacZ cells was observed, represented by fusiform cells in the central region of the rounded nucleus and without many irregularities in its support structure. Furthermore, as observed by the cytotoxicity tests, the treatment of cells with light at a fluence of 5 J/cm$^2$ does not trigger cell death and, therefore, does not produce morphological changes.

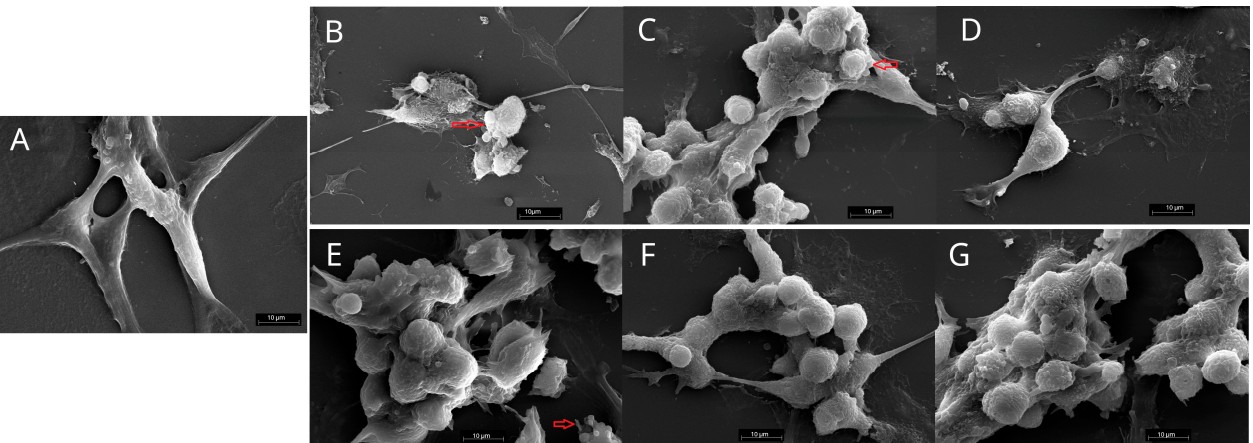

**Figure 5.** Micrograph obtained by scanning electron microscopy of 9 L/lacZ cells irradiated in an LED device at 660 nm, 5 J/cm$^2$, and 25 mW/cm$^2$, representing light control (**A**). Cells 9 L/lacZ submitted to PDT in the fluence of 5 J/cm$^2$ and 25 mW/cm$^2$ with 1.55 µg·mL$^{-1}$ FTC (**B**); 12.5 µg·mL$^{-1}$ FTC (**C**); 50 µg·mL$^{-1}$ FTC (**D**); 1.55 µg·mL$^{-1}$ PDZ (**E**); 12.5 µg·mL$^{-1}$ PDZ (**F**); and 50 µg·mL$^{-1}$ PDZ (**G**). Red arrow: signs of apoptosis; Magnification: 1.50 k×. Bar = 10 µM.

When these cells were submitted to PDT, it was observed that at all concentrations there was a change in morphology with the loss of the fusiform aspect, modified by cell stress and causing rounding of the cells, in addition to more significant irregularities in the support structure and evidence of apoptosis (Figure 5).

### 3.5. Analysis of HSP 27 and HPS 70 Protein Expression

When treated only with light (5 J/cm$^2$) or with the PS, an increase in HSP27 expression in the concentrations of 1.55 µg·mL$^{-1}$ and 12.5 µg·mL$^{-1}$ of both PS was observed, highlighting higher expression with the FTC. At the concentration of 50 µg mL$^{-1}$, there was a decrease in the expression of HSP27, both in the FTC and in the PDZ (Figure 6). When cells

were subjected to PDT with FTC, there was an increase in HSP27 expression at all tested concentrations, where we can observe that the lower the concentration, the more significant the increase in its expression. In the dark group, FTC concentrations of 1.55 $\mu$g·mL$^{-1}$ and 12.5 $\mu$g·mL$^{-1}$ showed a statistically significant difference with $p = 0.0041$ and $p = 0.0001$, respectively, compared with the control group. As well as in the irradiated groups, the same concentrations, when compared with the control group, showed a statistically significant difference with $p = 0.0034$ and $p = 0.0047$, respectively.

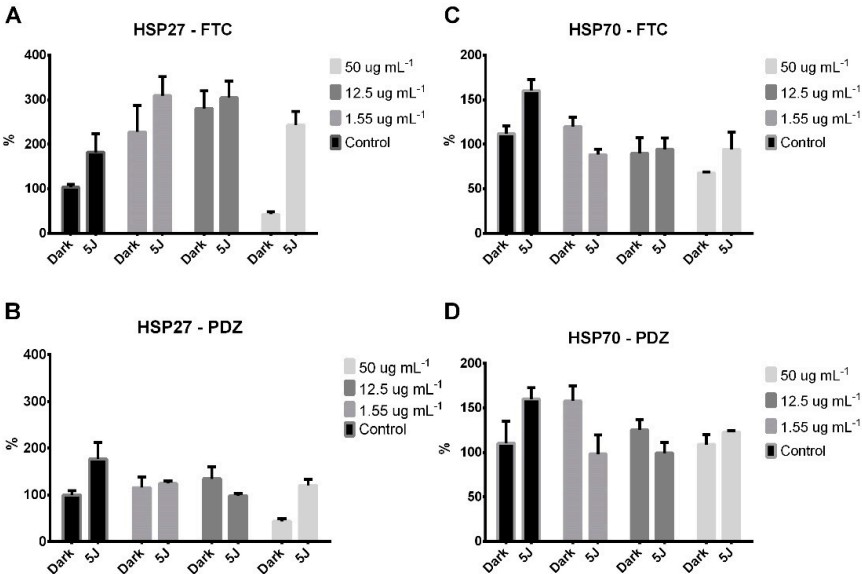

**Figure 6.** Analysis of HSP27 protein expression before and after PDT in the fluence of 5 J/cm$^2$. (**A**)—Expression of HSP27 in PDT with FTC at concentrations of 1.55 $\mu$g mL$^{-1}$, 12.5 $\mu$g mL$^{-1}$, and 50 $\mu$g mL$^{-1}$. (**B**)—Expression of HSP27 in PDT with PDZ at concentrations of 1.55 $\mu$g mL$^{-1}$, 12.5 $\mu$g mL$^{-1}$, and 50 $\mu$g mL$^{-1}$. Analysis of HSP70 protein expression before and after PDT in the fluence of 5 J/cm$^2$. (**C**)—Expression of HSP70 in PDT with FTC at concentrations of 1.55 $\mu$g mL$^{-1}$, 12.5 $\mu$g mL$^{-1}$, and 50 $\mu$g mL$^{-1}$. (**D**)—Expression of HSP70 in PDT with PDZ at concentrations of 1.55 $\mu$g mL$^{-1}$, 12.5 $\mu$g mL$^{-1}$, and 50 $\mu$g mL$^{-1}$.

In PDT with PDZ, there was a slight decrease in HSP27 expression. In the dark group, only the concentration of 50 $\mu$g mL$^{-1}$ of PDZ showed a statistically significant difference with $p = 0.0013$ compared to the control group. In the treated groups, the three concentrations showed statistical differences compared to the control group, with $p = 0.0029$ for the concentration of 1.55 $\mu$g mL$^{-1}$, $p < 0.0001$ for 12.5 $\mu$g mL$^{-1}$, and $p = 0.0013$ for 50 $\mu$g mL$^{-1}$.

As for HSP70 expression, it was observed that exposure to isolated light and isolated contact with PS at an FTC concentration of 1.55 $\mu$g mL$^{-1}$ and PDZ concentrations of 1.55 $\mu$g mL$^{-1}$ and 12.5 $\mu$g mL$^{-1}$ showed increased expression of HSP70. The other concentrations either showed a decrease in expression or maintained levels close to those expressed by the control group (Figure 6). In the application of PDT with FTC, it is observed that the concentrations showed a decrease in the expression of HSP70, with similar expression levels between the groups. Only the concentration of 50 $\mu$g mL$^{-1}$ shows a statistically significant difference with $p = 0.0029$ compared with the control, while in the PDT groups, the three concentrations showed statistical differences with $p = <0.0001$ for all concentrations.

In PDT with PDZ, the concentrations also showed a decrease in HSP70 expression. The concentration of 1.55 $\mu$g mL$^{-1}$ showed a statistically significant difference with $p = 0.0086$ compared with the control, while in the PDT groups, the three concentrations showed statistical differences with $p = 0.0029$ for the concentration of 1.55 $\mu$g mL$^{-1}$, $p = 0.0010$ for 12.5 $\mu$g mL$^{-1}$, and $p = 0.0419$ for 50 $\mu$g mL$^{-1}$.

### 3.6. Viability Analysis and Cell Adhesion of the L929 Cell Line after PDT

After observing the change in adhesion triggered by PDT with FTC and PDZ in cells of the 9 L/lacZ lineage, cell viability tests were carried out with fibroblasts representing normal cells to assess the effects caused in a common lineage. In the dark group, all concentrations tested maintained high cell viability, indicating that PS was not cytotoxic to fibroblasts. However, when subjected to PDT at all concentrations, cell viability was significantly reduced (Figure 7).

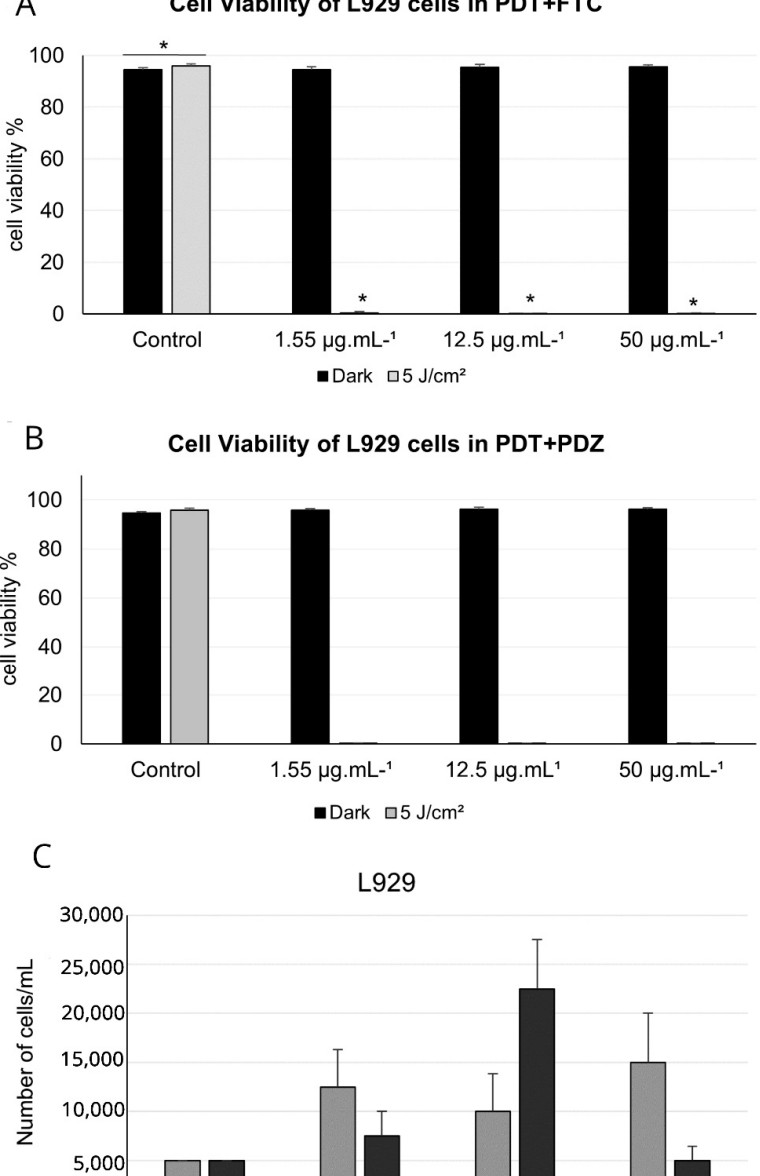

**Figure 7.** (**A**)—Viability of the L929 cells in the supernatant 18 h after PDT with FTC, and control group irradiated at 5 J/cm$^2$. (**B**)—Viability of the cells in the supernatant 18 h after PDT with PDZ, and control group irradiated at 5 J/cm$^2$. (**C**)—Number of cells present in the supernatant 18 h after PDT application, at a fluence of 5 J/cm$^2$, in L929 cells with FTC and PDZ, and control groups. Symbols for $p < 0.01$: *.

Regarding cell detachment after treatment, the same response pattern of 9 L/lacZ was not observed (Figure 7). There was a small number of cells in the supernatant, a result that corroborates the qualitative analysis observed during the Trypan Blue assay. Thus, the change in cell adhesion after PDT showed a different behavior between the tumor and regular cell lines.

## 4. Discussion

In continuation to the previous studies [3,12,13], the action of FTC and PDZ chlorine on GS cells was evaluated after a photodynamic process using low light fluences and PS concentrations. Seeking to understand the use of lower concentrations and fluences, aiming at future clinical application, low concentrations were chosen to compare the impact on viability, the interaction of PS with the cell, and morphological alterations triggered by the PDT. During the project, the impact of PDT on cell adhesion was observed, in a pattern inversely proportional to the concentration. To try to understand this event, measurements of heat shock proteins, involved in the cell adhesion process, were performed. Differences in the responses of both PS tested in this study were also observed.

The production of singlet oxygen is essential for Photodynamic Therapy's success and it was previously demonstrated that PDZ has a higher quantum yield than FTC, generating more $^1O2$. This behavior helps to understand the greater efficacy observed in the impact of PDT with PDZ on the viability of gliosarcoma cells than with FTC [15].

In all studied concentrations of PS, its accumulation in the cytoplasmic extension was observed without reaching the nuclear region. This fact had previously been described in previous studies by the group using higher concentrations of PS [3,12], in which it was possible to observe its location in the cytoplasm, as well as its co-location in organelles considered highly sensitive to oxidative damage, such as mitochondria and lysosomes [12].

The high accumulation capacity of PDZ in GS cells, even in lower concentrations, is related to its amphiphilic property and asymmetric polarity [12]. Thus, despite the apparent decrease in the fluorescence signal, the decrease in PS concentration did not represent a problem for its internalization in 9 L/lacZ cells. Therefore, the results presented in the cell viability test after PDT showed a significant reduction in the number of viable cells in all concentrations at the fluences of 5 J/cm$^2$ and 10 J/cm$^2$.

At the fluence of 1 J/cm$^2$, only the concentration of 1.55 µg mL$^{-1}$ did not show a high reduction in the number of viable cells, despite the mitochondrial activity of this group presenting a decrease.

The MTT assay evaluates a set of enzymatic activities related to cell metabolism [16]. The test relies on reducing the MTT salt to formazan crystals by NADPH-dependent enzymes in the mitochondria of viable cells. Several cellular activities stimulate the reduction of this salt, which may affect mitochondrial activity.

Thus, the decreased mitochondrial activity in cells treated with 1.55 µg mL$^{-1}$ of FTC and PDZ may occur due to a metabolic response to the PDT treatment, which was not enough to trigger cell death, as the cells remained viable, as observed in the Trypan Blue test. Therefore, the need to evaluate this in conjunction with the Trypan Blue exclusion assay helps understand the results obtained by evaluating the integrity of the cytoplasmic membrane.

According to Mosmann, 1983, the MTT assay is capable of detecting living cells, since they can convert the MTT salt into formazan crystals through reactions occurring mainly in active mitochondria; however, the test is not able to detect dead cells, so the result depends on the degree of cell activation and stimulation, measuring cell activation even without cell proliferation [17]. MTT is converted into mitochondria, which does not allow a single analysis of viability by this method, since Fotoenticine is present in mitochondria and, therefore, can lead to differences in the metabolic activity of this organelle. Stepanenko, 2015, published a study where it is pointed out that, depending on the cell type and experimental parameters studied, there may be discrepancies in the MTT test result, concluding that to avoid misinterpretation, supplementation of salt-based tetrazolium assays with other

non-metabolic assays is recommended [16]. For this reason, the viability test by exclusion with Trypan Blue was associated with the MTT test, to assess whether the metabolic alterations presented lead the cell to death, or if these cells could be in the process of recovery. The results can be interpreted as follows: although there is still mitochondrial activity in these cells, the Trypan test shows that they will not be able to recover, as the cell membrane has been damaged; therefore, these cells are in process of cell death. A previous study, using higher concentrations of PDZ and FTC, observed a predominance of apoptosis in the group treated with FTC, 87%, and 13% of cells in late apoptosis, while treatment with PDZ led to a predominance of cells in the process of late apoptosis, 76%, while 24% had necrosis markings [17].

Based on the results presented, the use of 5 J/cm$^2$ fluences was prioritized in the subsequent tests, considering the use of a lower light fluence. Although the Trypan Blue test is typically used to assess cell viability after treatment, a significant change in the number of adhered cells after PDT was observed throughout the experiment, especially at lower concentrations, thus, it quantitatively demonstrated cell detachment after therapy, which was present in more significant amounts at concentrations of 1.55 µg mL$^{-1}$ and 12.5 µg mL$^{-1}$, for both PS.

Despite this, the viability test of these cells showed more than 90% of dead cells, which may indicate that the loss of adhesion is a step corresponding to the process of cell death since several molecules and adhesion proteins located in the cell support structure can be affected in PDT, leading to cell damage and detachment [18]. Therefore, the use of these concentrations needs to be carefully evaluated to rule out the possibility that these cells pose a risk to the patient since the release of a viable cell could trigger the metastatic process, which presents a great problem, since GS is cancer with a high propensity for the development of extracranial metastases [19].

The analysis of cell morphology after PDT is an important step to be evaluated, mainly to identify possible characteristics related to this change in cell adhesion.

The 9 L/lacZ cells are spindle-shaped, with extensive cytoplasm and regular surfaces. When PDT is applied, we can observe from the images obtained by the SEM that the cells suffer great oxidative stress, assuming rounded shapes and more irregularities. These alterations were observed regardless of the concentrations and PS, reaching results similar to those observed in previous studies using the Giemsa method [3,12,13].

The same pattern of morphological alteration after PDT was observed in the studies by Ma and collaborators using Chlorine e6 in treating human colon cancer cells (SW480) at a concentration of 1 µg mL$^{-1}$, irradiated at the fluence of 6 J/cm$^2$. The authors reported cell atrophy and a marked decrease in the number of pseudopods and cells [20]. Therefore, consistency can be observed in the results presented by the chlorins, not only in the morphological changes but also in the reduction of cells after PDT, with approximately 50% of cell death at the concentration of 1 µg mL$^{-1}$ and 80% for the concentration of 8 µg mL.

In addition to changes in cell morphology, changes in adhesion may also be related to protein expression after PDT. In general, proteins are responsible for determining the shape and structure of cells and participating in vital processes.

One of the cellular responses to the external stressor stimulus is the production of heat shock proteins (HSP). These proteins are secreted, trying to protect cells from the processes triggered for the activation of cell death, acting in the synthesis, maturation, and repair of damaged proteins [21].

Oxidative stress triggered by PDT is the primary metabolic mechanism that causes damage to biological structures such as genetic material, proteins, and cell membranes. This process is also responsible for the increased concentration of some HSP in cells that have suffered oxidative stress during events such as modulation of autophagy and regulation of the inflammatory and immune response [22].

The non-linear relationship between the cellular stress process and Hsp release is complex and not yet fully understood. It is suggested that the amount of Hsp released may

vary according to the type and intensity of stress to which the cell is exposed, as well as according to the type of cell and the cellular environment in which it is found.

One of the proteins that actively participates in the redox process due to its modulation caused by the increase in ROS inside the cell is HSP70 [22].

The group of HSP70 proteins stands out due to its diversity of members participating in numerous cellular signaling pathways, including redox response pathways that act as chaperones helping in the folding of proteins formed during oxidative events and in pathways of inhibition of the cascade of apoptotic activation. Therefore, the protein synthesis of this group of proteins tends to be continuous due to their auxiliary metabolic functions and is modulated during stressful events [21,22].

Harmful events of great extensions, such as the accumulation of ROS triggered by PDT, can make the action of HSP70 inefficient, disrupting its cytoprotective function due to the extent of cell damage generated, enabling the triggering of cell death processes, stimulating the immune response and enabling control or eradication of target cells.

In the present study, it was shown that the production of HSP70 is continuous and can be modulated with the presence of photosensitizers or after PDT since the groups treated with the therapy had a decrease in their concentration, possibly due to the intense process of cell death evidenced in the cell viability and MTT assessments.

As Zhang and associates point out in their study, HSP70 undergoes alteration in its concentration and production due to the process of the response to oxidative events, so we can suggest that the accumulation of ROS generated by PDT is sufficient to cause an increase in the concentration of HSP, but it does not sustain the cytoprotective effect due to the extent of the damage generated in a short period, making the metabolic responses for cellular restoration and preservation inefficient [22,23].

Another prominent group among the HSPs is the low molecular weight HSPs such as the HSP27 group. These proteins act as a factor of resistance to apoptosis, capable of inhibiting the apoptotic pathway by increasing the intracellular antioxidant glutathione (GSH) by preventing the release of mitochondrial cytochrome c or by its direct binding to cytochrome c [23,24].

Among its characteristics, this group of proteins was identified as a metastatic marker for stomach, colon, and esophageal cancers, contributing to tumor invasion and resistance to PDT. Due to its characteristics, HSP27 presents an increased concentration in stressful events of great complexity that can lead to cell destruction mainly due to apoptotic processes. Thus, its intracellular increase can characterize a response of resistance to the stressor event [23–25].

Thus, after PDT in 9 L/lacZ cells, it was observed that the photosensitizer FTC induces a more intense cellular response when compared to PDZ, highlighting the concentrations 1.55 $\mu$g mL$^{-1}$ and 12.5 $\mu$g mL$^{-1}$, which showed increased expression of HSP27 with FTC, unlike PDZ. This increase suggests an attempt at cell adaptation and resistance to damage. Such a process may represent the emergence of resistant strains or even an increase in the spread of metastasis, corroborating the cell detachment seen in PDT with low concentrations or indicating the blockage of the apoptotic pathway. This result is similar to the work by Fontana et al. (2022), in which it was observed that the predominant death pathway in PDT with FTC in 9 L/lacZ cells is the necrotic pathway, followed by late apoptosis, performing the marking of propidium iodide and annexin in image cytometry [15].

Therefore, the use of FTC must be carefully analyzed since, for treating brain cancer, it is recommended to avoid the necrotic route due to the extension of the inflammatory effects and reduce the tumor's metastatic tendency. Thus, it is understood that the concentration of PS is just one of the factors that add to the stress mechanisms triggered by the presence of PS, change in the pH of the medium, increased formation of ROS, destruction of biomolecules such as proteins and membranes, and the interruption of mechanisms of signaling among other factors that are involved in the process of cellular stress; its variation is important but does not trigger a linear relationship in the face of damage generated and cellular responses.

According to the results presented in GT with the GS, we also sought to evaluate the behavior of the therapy in fibroblastic cells, characterizing the neighboring healthy tissue and observing whether cell adhesion would present the same alteration behavior.

At first, it was observed that SP did not alter the viability of L929 cells, an essential feature in ensuring safe and selective therapy. The selectivity of PDT occurs in two moments: (1) in the ability of normal cells to internalize PS and eliminate it faster than tumor cells [26] and (2) in the restriction of light emission only to the tumor area, in which the PS itself may show the boundary between diseased and healthy tissue through fluorescence [27]. Thus, only malignant cells will trigger the effects of PDT, minimizing the effects on healthy cells surrounding the lesion.

However, when PDT was applied to L929 cells, there was a significant reduction in viable cells in all tested concentrations, highlighting the importance of localized light emission.

As for cell detachment after 18 h of PDT, no change in the adhesion of these cells was observed. At all concentrations, the number of cells per well was tracked with the number prior to therapy application. The cell detachment observed after PDT in 9 L/lacZ cells was a lineage characteristic.

This result allows us to question which factors present in malignant cells can affect cell adhesion in such a way when consumed in PDT. HSP expression may represent one of the reasons. With this, we reinforce the need to deepen studies in this area to understand the processes that allow this response pattern and to what extent its benefits apply to therapy.

## 5. Conclusions

There was internalization of PS in the cytoplasmic extension of the 9 L/lacZ cell at all concentrations and the results of cytotoxicity tests showed an almost complete reduction of viable cells at fluences of 5 J/cm$^2$ and 10 J/cm$^2$. With this, the use of the lowest fluence was recommended, which presented significant results in all concentrations of the study. As a response to the effects of PDT, there was a change in the number of adhered cells after 18 h of treatment, morphological changes, and a change in the expression of heat shock proteins, in which the expression of the cytoprotective protein HSP70 decreased in both PS, while HSP27 showed an increase in expression in PDT with the FTC. Furthermore, chlorins were not cytotoxic for fibroblasts in the parameters used in this study, indicating the potential use of these PS in the treatment of brain cancer. We emphasize that the concentration of 50 μg mL$^{-1}$ at the fluence of 5 J/cm$^2$, of both PS, was the most advantageous for use in PDT in the treatment of gliosarcoma, interfering less in cell adhesion after 18 h of treatment. In general, FTC and PDZ showed similar results; however, due to the increase in HSP27 expression after irradiation of 9 L/lacZ cells with FTC, we can conclude that PDZ is more promising to be used in PDT, encouraging in vivo studies.

**Author Contributions:** Conceptualization, J.F.-S., G.d.S.V. and J.G.P.; methodology, J.F.-S., G.d.S.V., B.H.G. and J.G.P.; formal analysis, G.d.S.V. and B.H.G.; investigation, G.d.S.V. and B.H.G.; resources, J.F.-S.; data curation, G.d.S.V. and B.H.G.; writing—Original draft preparation, G.d.S.V. and B.H.G.; writing—Review and editing, J.F.-S., I.F., C.P.S. and J.G.P.; supervision, J.F.-S.; project administration, J.F.-S.; funding acquisition, J.F.-S. All authors have read and agreed to the published version of the manuscript.

**Funding:** This research was funded by São Paulo State Research Support Foundation (FAPESP), grant number 2016/12211-4. FINEP agreement 01.13.0275.00 and Coordination for the Improvement of Higher Education Personnel—Brazil (CAPES) Financing Code 001.

**Data Availability Statement:** The data will soon be made available in the institution's repository (repositorio.univap.br) in accordance with institutional policy.

**Acknowledgments:** The authors thank FAPESP/CEPOF-2013/07276–1.

**Conflicts of Interest:** The authors declare no conflict of interest.

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
