# Peer review of "Action of Photodynamic Therapy at Low Fluence in 9 L/lacZ Cells after Interaction with Chlorins"

_2673-7256, doi:10.3390/photochem3010006_

Round 1

Reviewer 1 Report

1)The figures need to be improved. For example, usage of a and b looks very confusing in Figure 1. In principle, the red emission from PS should be overlapping with cytoplasmic region so using two sets of arrows pointing towards different locations does not make sense. Figure caption is also problematic: for instance, please use "12.5" instead of "12,5" μg.mL-1 (please correct all similar issues throughout the manuscript). And consider put figure captions below the figures. Add clearer scale bars to the SEM images in Figure . Figure caption for Figure 3 is hard to understand (please add more details). 

2) What is the reason nucleus looks very different given different concentrations of PS?

3) The explanation for the different behavior caused by FTC and PDZ is also unclear. The authors cited existing papers but the discussion was repetitive and no much scientific insights (regarding the fundamental reason behind the chemical or physical reasons) were provided and no experimental proof was shown. This severely weakened the scientific soundness of the paper.

4) Overall the work looks too similar to the authors' existing papers and further clarity of the novelty should be emphasized.

Author Response

The authors thank the valorous contribution of both reviewers. We hope that the alterations carried in the manuscript could fit this manuscript to publication and respond to the questioning appropriately.

Reviewer 1:

1)The figures need to be improved. For example, the usage of a and b looks very confusing in Figure 1. In principle, the red emission from PS should be overlapping with cytoplasmic regions using two sets of arrows pointing towards different locations does not make sense. Figure caption is also problematic: for instance, please use "12.5" instead of "12,5" μg.mL-1 (please  correct all similar issues throughout the manuscript). And consider put figure captions below the figures. Add clearer scale bars to the SEM images in Figure . Figure caption for Figure 3 is hard to understand (please add more details).

Response: Alterations were performed throughout the manuscript. 

2) What is the reason nucleus looks very different given different concentrations of PS?

Response: All images were captured with a 63x magnification objective. However, in some cases, different zoom magnifications were used. For example, FTC 50; 12.5, and 1.55 were taken at 2.0 zoom; PDZ 50; 12.5, and 1.55 were performed with Zoom 1.8, 1.0, and 1.5, respectively. The images were mounted, and the bar was inserted in ZEISS Software ZEN. More information has been inserted in the figure caption.

3) The explanation for the different behavior caused by FTC and PDZ is also unclear. The authors cited existing papers but the discussion was repetitive and no much scientific insights (regarding the fundamental reason behind the chemical or physical reasons) were provided and no experimental proof was shown. This severely weakened the scientific soundness of the paper.

Response: In previous studies by the group, it was observed that PDT with PDZ and FTC at high concentrations reduced cell viability significantly. This study aimed to evaluate lower concentrations and fluences, aiming at future clinical application. However, when developing the experiments, it was observed that, compared with previous studies, the number of cells adhered to the treated wells reduced as the concentrations lowered. In this way, a  question arose if these cells would be dead when detached or still viable, becoming a problem for a possible clinical application. It was observed that with PDZ, more cells are released than with FTC, and the lower the concentration, the higher the number of cells released, for both PSs, although they appear to be all in the process of death. Thus, it was decided to test changes in heat shock protein expression to understand this process. Due to budgetary and equipment limitations, it was impossible to carry out additional tests. However, the article contributes essential information since it is proven that there is a change in the adhesion pattern of these cells. This pattern, if repeated in a biological environment, can cause problems such as the appearance of metastases, for example.   Glioblastoma is a very aggressive type of brain tumor that is difficult to treat, and metastasis is a big problem. If researchers understand how cancer cells attach to healthy tissue and spread throughout the body, they may be able to develop ways to prevent or inhibit this process. This study could help improve survival rates for patients with glioblastoma and other types of cancer. In addition, studying cell adhesion could help develop targeted therapies that target cancer cells and avoid affecting healthy cells. In summary, the study of cell adhesion is essential in treating glioblastoma and other brain tumors because it can help develop more effective and targeted treatments to prevent metastasis and improve patient survival rates.

4) Overall the work looks too similar to the authors' existing papers and further clarity of the novelty should be emphasized.

Response: In continuation to the previous studies, the action of FTC and PDZ chlorine on GS cells was evaluated after a photodynamic process using low light fluences and PS concentrations. Seeking to understand the use of lower concentrations and fluences for possible clinical application, low concentrations of PS were chosen to assess the impact on viability, the interaction of PS with the cell, and morphological changes. During the project, it was observed that the impact of PDT on cell adhesion was inversely proportional to the concentration. In order to understand this event, studies were carried out with the heat shock proteins involved in the cell adhesion process.

Reviewer 2 Report

In this work, the authors aimed to compare the effects of PDT using the photosensitizers (PSs) Fotoenticine (FTC) and Photo-dithazine (PDZ) at low concentrations and fluences. This study validated the PDT effect of PSs according to their impact on cell viability, the death process, and cellular regulatory mechanisms such as HSP expression and cell adhesion. However, the manuscript needs to be majorly improved and revised as following comments before publication.

1. In Fig 2, the authors only used different concentrations of PS with the same scale bar. Why the confocal images we see in the manuscript show different sizes of cells, especially nuclei?

2. In Fig 3A,3C or 3B, 3D, Trypan Blue and MTT are used for cytotoxicity evaluation. In principle, they should lead to similar results. Why do they show apparent difference at 1 J/cm² fluences, 1.55 µg/mL?

3. In Fig 3, the viability of cells treated with two PSs at 5 J/cm²,1.55µg/mL was almost zero, why is the number of cells different in the supernatant of the two groups in Fig 4A? Moreover, the higher the concentration of PS, the lower the cell viability in Fig 3. Why do we observe fewer cells in supernatant at the high concentration in Fig 4A?

4. In Fig 6, the relationship between the expression of heat shock proteins and the concentration of PSs is not linear. Why?

5. The full name of TFD should be provided.

6. The text and pictures need to be carefully checked and further enhanced (Figure labeling (case sensitive)).

Author Response

The authors thank the valorous contribution of both reviewers. We hope that the alterations carried in the manuscript could fit this manuscript to publication and respond to the questioning appropriately.

In this work, the authors aimed to compare the effects of PDT using the photosensitizers (PSs)Fotoenticine (FTC) and Photo-dithazine (PDZ) at low concentrations and fluences. This studyvalidated the PDT effect of PSs according to their impact on cell viability, the death process, andcellular regulatory mechanisms such as HSP expression and cell adhesion. However, themanuscript needs to be majorly improved and revised as following comments before publication.

  1. In Fig 2, the authors only used different concentrations of PS with the same scale bar. Why the confocal images we see in the manuscript show different sizes of cells, especially nuclei?

Response: All images were captured with a 63x magnification objective. However, in some cases different zoom magnifications were used. FTC 50; 12.5 and 1.55 were taken at 2.0 zoom; PDZ 50; 12.5 and 1.55 were performed with Zoom 1.8, 1.0 and 1.5, respectively. The images were mounted and the bar inserted in ZEISS Software ZEN. More information has been inserted in the figure caption.

  1. In Fig 3A,3C or 3B, 3D, Trypan Blue and MTT are used for cytotoxicity evaluation. In principle,they should lead to similar results. Why do they show apparent difference at 1 J/cm² fluences,1.55 μg/mL?

Response: According to Mosmann, 1983, the MTT assay can detect living cells, as they can convert the MTT salt into formazan crystals through reactions that occur mainly in active mitochondria; however, the test cannot detect cells. Dead, so the result depends on the degree of cell activation and stimulation, measuring cell activation even without cell proliferation. MTT is converted into mitochondria, which does not allow a single feasibility analysis by this method. FTC is present in mitochondria and, therefore, can lead to differences in the metabolic activity of this organelle. Stepanenko, 2015, published a study called "Pitfalls of the MTT assay: Direct and off-target effects of inhibitors can result in over/underestimation of cell viability," where is showed that, depending on the type of cell and parameters studied, there may be discrepancies in the result of the MTT test, concluding that, to falses, interpretation avoid, it is recommended to supplement the salt-based tetrazolium assays with other non-metabolic assays." Therefore, the viability test by exclusion with trypan blue is always used in our group, combined with the MTT test, to assess whether the metabolic alterations presented lead the cell to die or whether it is in the recovery process. The results can be interpreted as follows: Although there is still mitochondrial activity in these cells, the trypan test shows that they will not be able to recover, as the cell membrane has been damaged. These cells are, therefore, in the process of cell death.

Reference:

Mosmann T. Rapid colorimetric assay for cellular growth and survival: application to proliferation and cytotoxicity assays. J Immunol Methods. 1983 Dec 16;65(1-2):55-63.

Stepanenko, A.A., Dmitrenko, V.V., Pitfalls of the MTT assay: Direct and off-target effects of inhibitors can result in over/underestimation of cell viability, Gene (2015), doi: 10.1016/j.gene.2015.08.009

  1. In Fig 3, the viability of cells treated with two PSs at 5 J/cm²,1.55μg/mL was almost zero, why is the number of cells different in the supernatant of the two groups in Fig 4A? Moreover, the higher the concentration of PS, the lower the cell viability in Fig 3. Why do we observe fewer cells in supernatant at the high concentration in Fig 4A?

Response: In previous studies by the group, it was observed that PDT with PDZ and FTC at high concentrations could significantly reduce cell viability. This study aimed to evaluate lower concentrations and fluences, aiming at future clinical application. However, when developing the experiments, it was observed that, in comparison with previous studies, the number of observed cells adhered to the treated wells reduced the lower the concentration used. In this way, a new question arose, would these cells be dead, and if released, would they still be viable, becoming a problem for a possible clinical application. It was observed that with PDZ, more cells are released than with FTC, and the lower the concentration, the greater the number of cells released for both PSs. Thus, it was decided to test changes in heat shock proteins to understand this observed process.

  1. In Fig 6, the relationship between the expression of heat shock proteins and the concentratin of PSs is not linear. Why?

Response: For example, some studies suggest that the amount of Hsp released may vary according to the type and intensity of stress the cell is exposed to, the type of cell, and the cellular environment in which it is found.

Furthermore, the activation of Hsp production can be regulated by intracellular mechanisms, such as the activation of signal transduction signals and specific transcription factors, suggesting that Hsp release is a dynamic and regulated process that several factors can influence.

Thus, it is understood that the concentration of PS is just one of the factors that add to the stress mechanisms triggered by the presence of PS, change in the pH of the medium, increased formation of ROS, destruction of biomolecules such as proteins and membranes, interruption of mechanisms of signaling among other factors that are involved in the process of cellular stress, its variation is essential. Still, it does not trigger a linear relationship due to damage generated and cellular responses.

In summary, the non-linear relationship between the cellular stress process and Hsp release is complex and needs to be fully understood. However, Hsp plays an essential role in protecting cells against the harmful effects of stress and maintaining cellular homeostasis.

  1. The full name of TFD should be provided.

Response: We appreciate the reviewer's attention. Unfortunately, TFD is the abbreviation of Photodynamic therapy in Portuguese. The change was made throughout the text

  1. The text and pictures need to be carefully checked and further enhanced (Figure labeling(case sensitive))

Response: The authors hope that the changes made throughout the manuscript have made it more precise and suitable for publication.

Round 2

Reviewer 1 Report

The authors have addressed my previous comments.